# Body Composition and Anthropometric Indicators in Children and Adolescents 6–15 Years Old

**DOI:** 10.3390/ijerph191811591

**Published:** 2022-09-14

**Authors:** Milena Kobylińska, Katarzyna Antosik, Agnieszka Decyk, Katarzyna Kurowska, Diana Skiba

**Affiliations:** Department of Medical Sciences and Health Sciences, Institute of Health Sciences, Siedlce University of Natural Sciences and Humanities, 08-110 Siedlce, Poland

**Keywords:** overweight, obesity, nutritional status, children, adolescents

## Abstract

The problem of overweight and obesity among children and adolescents has now become a major public health challenge worldwide. The aim of this study was to assess the effect of age and gender on body composition components and anthropometric indices of children and adolescents aged 6–15 years; in addition, the study aimed to assess body composition indices in relation to BMI (Body Mass Index) levels. The study was conducted at the end of 2019 and in the first quarter of 2020 among 181 pupils attending primary schools. Waist circumference, hip circumference, body weight, and body height were measured. The collected data were used to calculate and then to analyse BMI, WHR (Waist-Hip Ratio), and WHtR (Waist to Height Ratio) indices. Body composition was determined with the use of the Bioelectrical Impedance Analysis (BIA). The analyses’ statistics were performed using IBM SPSS Statistics 24 and Excel. The statistical methods used included Chi^2^ tests of independence, one-factor analysis of variance, and two-factor analysis of variance taking into account the level of α = 0.05. Based on the results, there were no statistically significant differences in the gender and age distributions of the BMI groups. However, the analysis of interaction effects confirmed that there were statistically significant differences according to pubertal age and gender in body fat, muscle mass, hydration, and WHR. Body composition in boys and girls before the age of 12 is similar while, after the age of 12, there are differences between boys and girls in terms of body composition so there is a need to deepen the assessment of body mass, especially in adolescents at the age of puberty, by body composition analysis using the BIA method.

## 1. Introduction

The problem of overweight and obesity among children and adolescents is now becoming a growing health problem worldwide [1,2]. According to a recent World Health Organization (WHO) report, 340 million children and adolescents between 5 and 19 years of age and nearly 40 million children under 5 years of age were overweight or obese [3]. Excessive weight gain during childhood and adolescence is associated with an increased probability of obesity and also premature death in adulthood [4]. Obesity acquired in childhood not only increases the probability of obesity in adulthood, but is also closely related with the risk of developing both cardiovascular disease and diabetes [5,6]. Health consequences may still persist despite the normalisation of body weight in a person who was burdened with obesity in childhood. Due to the number of health consequences of overweight and obesity, it is essential to implement prevention as early as possible. Health education for children and adolescents is crucial, as well as the cooperation of the school and family environment in this matter [7,8].

During the development and puberty of children and adolescents, it is very essential to assess the nutritional status. It provides an early detection of disorders and their correction [3]. The anthropometric methods used to assess nutritional status include measurements of body weight, body height, circumferences (waist and hips), and also indices such as BMI (Body Mass Index), WHR (Waist-Hip Ratio), and WHtR (Waist to Height Ratio) are used. In paediatric practice, body weight, body height and BMI are assessed using percentile grids for age and gender [9]. The BMI, which is commonly used to categorize body weight, unfortunately has its limitations, as it does not provide information about the content of individual components of body composition, such as the content and distribution of body fat, muscle mass, and body water [10,11]. To improve the monitoring of body weight, it is recommended to use Bioelectrical Impedance Analysis (BIA), which is characterized by high accuracy and repeatability, is also used for body composition analysis [12,13]. This method allows a non-invasive, fast and complex assessment of the content of individual body components, i.e., muscle tissue, adipose tissue and body water [14]. This is important for adolescents in the stage of puberty proper, during which there are intensive changes in the endocrine system, rapid growth, as well as the development of secondary and tertiary sexual characteristics, which ultimately affects the formation of morphological and functional characteristics of the body [15]. The assessment of body composition using BIA analysers allows for a wider range of anthropometric studies and increases the usefulness of the results obtained in educational and intervention practice [16], which justifies undertaking research in this area. The aim of this study was to assess the effect of age and gender on body composition components and anthropometric indices of children and adolescents aged 6–15 years; in addition, the study aimed to assess body composition indices in relation to BMI levels.

## 2. Materials and Methods

### 2.1. Participants

The study was conducted at the end of 2019 and in 2020, from January to March, among 181 primary school pupils from the Mazovia Province in Poland, between 6 and 15 years of age: 96 girls and 85 boys.

The study could only be carried out for two and a half months in 2020 due to the COVID-19 pandemic.

### 2.2. Measurement

Waist circumference, hip circumference, body weight, and body height were measured. Based on the results, BMI, WHR, and WHtR were calculated. Body composition was measured using BIA. During each on-site physical examination, a special person was appointed to conduct on-site supervision to ensure the accuracy and standardization of the measurement methods and records of each measurement indicator. Measurements were taken in the morning, fasting, after emptying the bladder and after a 12 h break from intense physical activity. Subjects were informed in advance to avoid caffeine consumption and smoking before the measurement. In girls, the procedure was performed on the 10th to 15th day of the menstrual cycle.

#### 2.2.1. Waist Circumference

Waist circumference was measured using a Seca 201 anthropometric tape, in underwear, in a standing position with feet together, with body weight evenly distributed on both feet and with the bladder emptied. After several natural breaths of the subject, a measurement was taken in a position parallel to the floor level, halfway between the lower edge of the last rib arch and the highest point of the iliac crest. The measurement was repeated three times and the average of the measurements was calculated.

#### 2.2.2. Hip Circumference

Hip circumference was also measured using a Seca 201 anthropometric tape, in underwear, in a standing position with feet together, with body weight evenly distributed on both feet. This measurement was carried out at the point of greatest girth of the buttocks below the hip plates, in a direction parallel to the floor. The measurement was repeated three times and the average of the measurements was calculated.

#### 2.2.3. Body Height

Body height was measured using a Seca 213 stadiometer. The subject stood barefoot in a freely upright position, with his back to the height gauge, heels joined and feet slightly angled apart, with his hands placed loosely along his body. The buttocks, heels, shoulder blades and occiput were adjacent to the plane of the device. The head, on the other hand, was placed in the Frankfurt position (the upper edges of the external auditory orifices and the lower edge of the orbit were at the same level—the ear canal was in line with the cheekbone). Body height was measured without shoes, headgear, elaborate hairstyles and hair ornaments. The measurement was repeated three times, and the average of the measurements was calculated.

#### 2.2.4. Waist-Hip Ratio (WHR)

Based on anthropometric measurements—waist circumference (cm) and hip circumference (cm), the WHR > 0.8 for female gender and WHR > 1 for male gender were used as criteria for abdominal obesity [17].

#### 2.2.5. Body Mass Index (BMI)

It was calculated according to the formula: body weight (in kg)/body height (in meters)^2^. BMI was assessed based on the Polish OLA and OLAF percentile grids, and the following interpretation of the results was made:

Underweight < 5 percentile;

Thinness above 5 below 25 percentiles;

Normal weight above 25 below 85 percentiles;

Overweight above 85 below 95 percentiles;

Obesity ≥ 95 percentiles [18].

#### 2.2.6. Waist to Height Ratio (WHtR)

This index was calculated by dividing waist circumference (cm) by body height (cm).

We evaluated the prevalence of abdominal obesity in children and adolescents using WHtR ≥ 5.

#### 2.2.7. Bioelectrical Impedance Analysis (BIA)

Body composition was determined by BIA using a Tanita DC-430 S MA analyser. Body fat, lean body mass and body water were measured. According to the producer’s recommendations, the test subjects wore only underwear. During the analysis, it was made sure that the test person had bare, clean feet that that are properly positioned on the platform and are in contact with the electrodes.

The study was conducted in accordance with the Declaration of Helsinki and was approved by the Ethics Committee (No. 2/2017, No. 11/2019). The study was carried out within the framework of the research project “Food and nutrition versus human health”, and it was conducted after written consent of the parents of the study participants was obtained.

### 2.3. Data Analysis

To verify the research hypotheses, statistical analyses were performed using IBM SPSS Statistics 24 and Excel. Analyses were carried out on a sample of 181 observations, which allowed detection of moderate difference effects with the assumption of division into three to five BMI groups at a test power of 1-β > 0.80 and a significance level of α = 0.05. Given the resulting distribution of BMI in the sample, it was divided into three levels (underweight, normal, overweight), while the extreme groups were included in underweight and overweight, respectively, as they accounted for fewer than 10 observations, which would translate into unrepresentative analysis results. The statistical methods used included chi-square to analyse differences in exceeding the critical WHR and WHtR values, one-way ANOVA to verify differences in body composition components according to BMI, two-way ANOVA to trace the interaction between gender and age in the variation of body composition components, which were continuous variables. In the case of one-way ANOVA, a Kruskal–Wallis H analysis was additionally applied to control for the disparity effect of samples with different BMI levels. In addition, in each analysis of variance, a Bonferroni correction was applied to the significance threshold for pairwise comparisons.

## 3. Results

### 3.1. Characteristics of Respondents

A total of 181 children between 6 and 15 years of age participated in the study, including 96 girls (53%) and 85 boys (47%). Nearly 50% of the students who participated in the study were between the ages of 7 and 9 years. More than 27% of the study population were children aged 10 to 12 years, and almost 15% of all pupils were in the age range of 13–15 years. The smallest percentage (8.29%) of the study group were children under 7 years old. In the study group, the average weight among the students was 36.22 ± 13.11 kg. The mean value of body height was 142.18 ± 16.07 cm. The average waist circumference value among the children was 60.53 ± 8.03 cm, while the average hip circumference was 74.46 ± 10.65 cm. The mean BMI in the study group was 17.27 ± 2.94 kg/m^2^. Based on this research, the mean WHR was 0.81 ± 0.05, while the mean WHtR in the studied group of children was 0.42 ± 0.04 (Table 1).

In this study, normal body weight was found in 57.5% of students (*n* = 104), overweight in 21.5% (*n* = 39), and underweight in 21.0% (*n* = 38).

### 3.2. BMI Values Taking into Account Gender and Age

Based on the observation of the results of our own study, we found that there were no statistically significant differences in the gender and age distributions in each BMI group (Table 2).

### 3.3. Occurrence of Abdominal Obesity in the Study Group

The analyses conducted showed that there were no statistically significant differences between individuals with different BMI levels in terms of WHR. Observation of the values of the ratio of excesses of the index showed variations between 23.7% and 32.7% within all groups, suggesting that about a quarter of the students exceeded the index regardless of weight and height (Table 3).

Statistically significant differences were found for WHtR (Table 3), which indicated that overweight subjects were nearly three times more likely to exceed the critical value (20.5%) compared to normal weight subjects (6.7%) and nearly eight times more likely to exceed the critical value compared to underweight subjects (2.6%). Pairwise comparison tests with Bonferroni correction confirmed that only WHtR excesses in overweight subjects were significantly more frequent compared to underweight and normal BMI subjects. In contrast, there were no differences between the number of exceeded index values in underweight and normal weight subjects.

### 3.4. Body Composition Components including BMI

Performed analyses of variance (Table 4), confirmed statistically significant differences across all body composition indices according to BMI level. Analysis of the differences between pairwise means using the Bonferroni correction confirmed that, with the exception of WHR, there were statistically significant differences between all groups at the *p* < 0.001 level. In terms of WHR, we found that underweight subjects had a significantly lower WHR than overweight subjects (*p* = 0.015), but there were no differences between normal BMI subjects compared to underweight (*p* = 0.531) and overweight (*p* = 0.097) subjects. However, it was confirmed that as BMI increased, there was an increase in body fat, waist and hip circumference, and an increase in WHtR—these indices were highest among overweight individuals. In contrast, a lower BMI promoted an increase in muscle tissue and hydration, both of which were highest among the underweight subjects. Depending on the index, BMI explained between 5% and 51% of the variance in the results of the collected data. It turned out that BMI best explained the increase in waist circumference, while WHR was the weakest (Table 4).

This was followed by an analysis of the effect of pubertal age and gender on the various components of body composition in the sample. The analysis was performed using a two-factor analysis of variance for the entire study sample due to the test’s too low power (1 − β < 0.80) for individual BMI groups. The results are shown in Table 5. The analysis of interaction effects (Table 5) confirmed that there are statistically significant differences according to pubertal age and gender, which explain between 5% and 8% of the variance in the data in terms of body fat, muscle mass, body hydration and also WHR index. Analyses of pairwise comparisons with Bonferroni’s correction for the significance threshold, confirmed that there were no statistically significant differences in all of the above body composition components among boys and girls under 12 years of age. In turn, statistically significant differences between boys and girls at puberty (12–15 years) were confirmed. It turned out that pubertal girls were characterized by higher body fat, lower muscle, lower body hydration and lower WHR compared to boys at this age.

The performed analyses of variance for the main effect of age (Table 5) confirmed a statistically significant effect of age on waist and hip circumference, as well as WHR and WHtR indices. They found that waist circumference was smaller among children under 12 years old (58.63 ± 0.60) compared to those aged 12–15 (67.17 ± 1.09). Likewise for hip circumference, which was on average larger in older children (87.50 ± 1.21) than younger children (70.65 ± 0.67). On the other hand, WHR and WHtR were, on average, higher under 12 years of age compared to children aged 12–15: WHR (0.83 ± 0.01 vs. 0.77 ± 0.01), WHtR (0.43 ± 0.01 vs. 0.41 ± 0.01). Analysis of the proportion of explained variance indicated that age explained the most in terms of hip width (46%), while it explained the least in terms of WHtR change (7%).

The main effect of gender was also confirmed in terms of body fat indices, muscle mass, body hydration, waist circumference and WHR (Table 5). It turned out that boys had lower body fat (14.00 ± 0.75) compared to girls (19.42 ± 0.73). They also found greater muscle mass and better hydration among boys compared to girls, respectively: muscle mass (81.50 ± 0.70 vs. 76.43 ± 0.69), and hydration (62.98 ± 0.54 vs. 58.99 ± 0.53).

In addition, boys on average had a larger waist circumference (64.53 ± 0.89) compared to girls (61.27 ± 0.87), as well as a higher WHR (0.81 ± 0.01 vs. 0.79 ± 0.01). The results explained between 5% and 14% of the variance in the data.

Summarizing these results, age and gender did not differentiate BMI values. Adipose tissue, waist circumference, hip circumference and WHtR increased with increasing BMI, while lower BMI promoted an increase in muscle tissue and hydration in the study group. Analyses showed that there were no statistically significant differences between subjects with different BMI levels in terms of WHR, while statistically significant differences were found in terms of WHtR. Overweight subjects exceeded the critical WHtR value significantly more often than underweight and normal-weight subjects. Our results confirmed statistically significant differences in all body composition indices according to BMI level. Our own analysis showed the influence of pubertal age and gender on body composition components. Significant differences in body composition, between pubertal boys and girls, were confirmed. Girls were characterized by more body fat, less muscle, less body hydration and lower WHR compared to boys at this age (12–15 years). Among boys and girls under the age of 12, there were no statistically significant differences in all components of body composition.

## 4. Discussion

To assess nutritional status, the most commonly used index in various studies was BMI (73%), which was related to percentile grids according to age and sex [19,20]. Percentile grids are most commonly used in paediatric practice with the assessment of body weight and height [9]. In Poland and abroad, different criteria have been adopted for the assessment of nutritional status results, therefore difficulties occur comparing the results with other authors [21].

In our study, normal weight was found in 57.5% of students (*n* = 104), overweight in 21.5% (*n* = 39), and underweight in 21.0% (*n* = 38). While the study by Sztander et al. [22] showed that 69.75% of children had a normal body weight, 14.4% underweight, 14.81% overweight, and obesity occurred in 1.02% of the respondents, whereas, in the study by Pacian et al. [23], it was shown that in the studied group of pupils aged 6 to 13 years old, 38.0% of children were overweight, 23.1% were underweight, and the percentage of children with normal body weight was 38.9%, which was almost 1.5 times less in relation to this study. Baran [24] in his study showed that the percentage of children with overweight was more than two times lower (9.4%) than in this study.

An alarming phenomenon among children is malnutrition, which can lead to developmental disorders, deterioration of mental and physical skills, disturbed metabolic processes of macronutrients, anaemia, decreased immunity and hormonal disorders [25]. In the study by Więch et al. [26], in terms of BMI in relation to percentile grids, it was found that none of the children had body mass deficiency, which is not confirmed by this study. The authors’ own research did not show a statistically significant relationship between gender and BMI, which was confirmed by the studies by Wyka et al. [27].

The authors of the study also did not demonstrate a statistically significant relationship between gender and normal body weight of the subjects. A higher prevalence of obesity among boys compared to girls was found in Greece and the USA [28,29]. Głowacka et al. [30] confirmed this conclusion, observing in their study that obesity was more common in boys.

In this study, no statistical significance difference was observed between age and BMI values, which is confirmed by the study by Sztander et al. [22], who also did not demonstrate a significant relationship between BMI categories—overweight and obesity, normal and underweight—and age.

In this study, no statistical significance difference was observed between WHR and BMI. However, it was observed that the largest number of participants with abdominal obesity were in the BMI group with normal body weight (*n* = 34), which was 18.8% of the whole study group. The study by Malczyk et al. [9] proved that 78% of girls and all boys with excessive body weight did not have abdominal obesity based on the calculated WHR, which was confirmed by this study. It was shown that certain people with normal body weight have metabolic abnormalities that occur in people with abdominal obesity. They have been described as metabolically obese normal-weight (MONW). Despite attempts to accurately characterise MONW individuals, there is still no generally accepted definition and criteria for the diagnosis of this syndrome. Studies have shown that the global prevalence of MONW ranges from 5 to 45%. Thus far, the prevalence of MONW has not been definitively studied and it is difficult to determine the real scale of the described phenomenon. Firstly, because different researchers have used different criteria and methods to classify MONW. Furthermore, studies have been conducted on diverse ethnic and age groups ranging from adolescents to adults. It has been reported that abdominal obesity is noted in a subset of children without excess body weight [31,32,33,34]. In a study of adolescents from Maracaibo, Venezuela, conducted by Molero-Conejo et al. [35], the prevalence of MONW was estimated at 37%. There are no studies in Polish literature on the prevalence of individuals with MONW.

Based on the WHtR index, abdominal obesity was observed in only 8.84%, while 91.16% of the studied students had no abdominal obesity. Krajewska et al. [36], based on WHtR index, observed abdominal obesity in 98.9% of the studied girls and also in 98% of the studied boys. On the other hand, Malczyk et al. [37] in their study showed that abdominal obesity was not observed in the majority of children (92%), which was confirmed by the results of our own research. Scientists note that regardless of sexual maturation, there is evidence that boys may exhibit higher levels of body fat in the waist region. Puberty may have an impact on increased fat content in the central region, which may lead to a more android shape in boys [38]. In this study, a statistically significant relationship was observed between WHtR and BMI. WHtR excesses in overweight subjects were significantly more frequent compared to underweight and normal BMI subjects. Błaszczyk-Bębenek et al. [21] showed that the majority of the examined students were not at risk of abdominal obesity (92.8%) referring to the WHtR index, which is confirmed by this study.

It was confirmed that as BMI increased, there was an increase in body fat, waist and hip circumference, and an increase in WHtR—these indices were highest among overweight individuals. Similar conclusions were reached by Gołąbek and Majcher [39], who demonstrated in their study that adipose tissue content increases with increasing BMI. In the study by Więch et al. [26], it was found that the adipose tissue content was significantly higher in overweight and obese children compared to children with normal body weight, which is confirmed by this study.

The authors’ own research showed that lower BMI was conducive to the growth of muscle tissue and hydration, both of which were the highest among underweight people. There are no studies in the literature on the correlation between BMI and average % muscle mass in the age group of 6–15 years, so it is difficult to discuss this issue. This raises the need for further research in this direction.

Based on the results of our study, we found that overweight children significantly had the lowest percentage of body water compared to underweight and normal weight students. The average body water percentage in the 181 examined children was 61%. Błaszczyk-Bębenek et al. [21] showed that the majority of children (72.3%) had body water deficiency—below 65%, which is confirmed by this study. There are few published studies on quantitative comparison of total body water (TBW) for age, body weight, height or body surface area in normal and overweight children [40]. The lower body water content related to obesity is a result of higher body fat and lower lean body mass [41].

This study showed that people with excessive weight subjects had significantly higher waist circumference than normal weight and underweight subjects. Analysis showed that waist circumference increased with increasing body weight. The average waist circumference among the 181 students studied was 60.53 ± 8.03 cm. The increase in central obesity in both girls and boys does not appear to be an isolated trend. There is evidence that abdominal obesity is increasing faster than obesity in general (assessed on the basis of BMI) [42].

Our own research showed that girls in adolescence were characterised by higher body fat, lower muscle tissue, lower body hydration and lower WHR compared to boys in adolescence, which is also confirmed by the results obtained by Burdukiewicz et al. [43].

During puberty, men gain more fat-free mass, while women acquire significantly more fat mass. Sexual dimorphism occurs to a small extent at birth, but significant differences appear during puberty. The development of this dimorphism in body structure is largely regulated by endocrine factors [44].

Despite showing significant trends, the study presented has some limitations. The study only lasted for a period of 3 months, which limited the size of the study population. It was planned to continue the study in this age group of children and adolescents. However, the outbreak of the COVID-19 pandemic and related limitations, including school closures and including online education, caused the study to be discontinued. Another limitation is that body composition is influenced by a variety of factors, such as dietary behaviour, physical activity level and socioeconomic status, among others, so it is advisable to expand the study to include the aforementioned factors.

A strength of the study was that anthropometric measurements and body composition analysis were performed by one person, who was supervised by a second specialist in the field, thus avoiding measurement errors. Anthropometric measurements and body composition analysis were performed using the same diagnostic equipment, which increased the reliability of the study’s results. In addition, the analysis included only children and adolescents belonging to the same ethnic group.

## 5. Conclusions

Age and gender are not variables that differentiate BMI values.

As BMI increased, there was an increase in body fat, waist and hip circumference, and an increase in WHtR, while a lower BMI promoted an increase in muscle tissue and hydration in the study group.

Body composition in boys and girls before the age of 12 is similar while, after the age of 12, there are differences between boys and girls in terms of body composition. Gender differences in body composition are manifested by a higher proportion of muscle mass in boys and higher body fat in girls, so there is a need to deepen the assessment of body mass, especially in adolescents in adolescence, with BIA body composition analysis, which, unlike BMI, provides detailed information about the individual components of body composition, including the amount of body fat, muscle mass and body water.

## Figures and Tables

**Table 1 ijerph-19-11591-t001:** Characteristics of the studied population including body weight, body height, BMI, waist circumference, hip circumference, WHR and WHtR.

Variables	N	M	Min	Max	SD
**Weight (kg)**	181	36.22	17.60	81.20	13.11
**Height (cm)**	181	142.18	109.00	186.50	16.07
**BMI (kg/m^2^)**	181	17.27	12.50	27.30	2.94
**Waist circumference (cm)**	181	60.53	44.00	83.00	8.03
**Hip circumference (cm)**	181	74.46	56.00	104.00	10.65
**WHR**	181	0.81	0.69	0.95	0.05
**WHtR**	181	0.42	0.33	0.59	0.04

N—number of participants; SD—standard deviation; M—medium; BMI—Body Mass Index; WHR —Waist-Hip Ratio; WHtR—Waist to Height Ratio.

**Table 2 ijerph-19-11591-t002:** BMI values taking into account age and gender.

Variables	Interpretation	Underweight		Normal		Overweight		χ^2^	*df*	*p*	*V*
		N	%	N	%	N	%				
**Age**	<7 years	7	18.4%	7	6.7%	1	2.6%	11.28	6	0.080	0.18
	7–9 years	19	50.0%	49	47.2%	22	56.4%				
	10–12 years	8	21.1%	28	26.9%	13	33.3%				
	13–15 years	4	10.5%	20	19.2%	3	7.7%				
**Gender**	Male	15	39.5%	47	45.2%	23	59.0%	3.25	2	0.197	0.13
	Female	23	60.5%	57	54.8%	16	41.0%				

N—number of participants; χ^2^—chi-square test; *df*—number of degrees of freedom; *p*—significance of differences test; *V*—Kramer’s V strength index.

**Table 3 ijerph-19-11591-t003:** Interpretation of the WHR and WHtR index taking into account the BMI.

Index	Interpretation	Underweight		Normal		Overweight		χ^2^	*df*	*p*	*V*
		N	%	N	%	N	%				
**WHR**	No abdominal obesity	29	76.3%	70	67.3%	29	74.4%	1.41	2	0.494	0.09
	Abdominal obesity	9	23.7%	34	32.7%	10	25.6%				
**WHtR**	No abdominal obesity	37	97.4%	97	93.3%	31	79.5%	8.99	2	0.011	0.22
	Abdominal obesity	1	2.6%	7	6.7%	8	20.5%				

N—number of participants; χ^2^—chi-square test; *df*—number of degrees of freedom; *p*—significance of differences test; *V*—Kramer’s V strength index.

**Table 4 ijerph-19-11591-t004:** Analysis of differences using one-way analysis of variance comparing body composition components among individuals with different BMI levels.

Dependent Variables	Underweight (*n* = 38)		Normal (*n* = 104)		Overweight (*n* = 39)		*F*(2,178)	*p*	*p*-Rank	η^2^
	M	SD	M	SD	M	SD				
**Adipose tissue (%)**	10.61	2.76	16.25	4.57	23.53	6.36	72.82	<0.001	<0.001	0.45
**Muscle Mass (%)**	84.53	2.66	79.42	4.37	72.48	5.98	70.43	<0.001	<0.001	0.44
**Water content (%)**	65.38	2.04	61.33	3.35	56.01	4.65	71.55	<0.001	<0.001	0.45
**Waist circumference (cm)**	52.05	3.85	60.24	6.01	69.59	6.07	93.24	<0.001	<0.001	0.51
**Hip circumference (cm)**	65.11	6.50	74.45	9.17	83.64	9.70	42.63	<0.001	<0.001	0.32
**WHR**	0.80	0.04	0.81	0.05	0.84	0.06	4.45	0.013	0.023	0.05
**WHtR**	0.39	0.03	0.42	0.03	0.48	0.04	69.88	<0.001	<0.001	0.44

*p*—significance of differences test; *p*-rank—non-parametric Kruskal–Wallis H test; η^2^—proportion of variance explained by BMI; M—medium; SD—standard deviation.

**Table 5 ijerph-19-11591-t005:** Analysis of the interaction effect of pubertal age and gender on body composition components.

Dependent Variables	Male <12 Years (*n* = 64)		Female <12 Years (*n* = 75)		Male 12–15 Years (*n* = 21)		Female 12–15 Years (*n* = 21)		Age Effect		Gender Effect		Interaction Effect	
	M	SD	M	SD	M	SD	M	SD	*p*	η^2^	*p*	η^2^	*p*	η^2^
**Adipose tissue (%)**	15.81	4.67	17.18	6.85	12.19	4.75	21.67	6.75	0.680	<0.01	<0.001	0.13	<0.001	0.08
**Muscle Mass (%)**	79.64	4.40	78.53	6.45	83.36	4.49	74.33	6.38	0.808	<0.01	<0.001	0.13	<0.001	0.08
**Water content (%)**	61.62	3.41	60.64	4.98	64.35	3.45	57.34	4.93	0.705	<0.01	<0.001	0.14	<0.001	0.08
**Waist circumference (cm)**	59.97	7.32	57.31	7.11	69.10	5.30	65.24	7.50	<0.001	0.21	0.009	0.04	0.631	0.00
**Hip circumference (cm)**	72.17	7.72	69.13	7.93	87.33	6.32	87.67	8.96	<0.001	0.46	0.327	0.01	0.222	0.01
**WHR**	0.83	0.04	0.83	0.05	0.79	0.04	0.74	0.04	<0.001	0.25	0.003	0.05	0.004	0.05
**WHtR**	0.44	0.04	0.43	0.05	0.41	0.03	0.40	0.05	<0.001	0.07	0.160	0.01	0.981	0.00

η^2^—proportion of variance explained by BMI; *p*—significance of differences test; M—medium; SD—standard deviation.

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
