# Peer review of "Body Composition and Anthropometric Indicators in Children and Adolescents 6–15 Years Old"

_ijerph, 2022, doi:10.3390/ijerph191811591_

Round 1

Reviewer 1 Report

Thank you for your interesting manuscript! However, I have some suggestions for the clarification and the better one.

Major comments

1.      Concerning the methodology, how did you calculate the sample size? As the total sample size was 181, how did you manage the consequences of smaller sample size for the validity and power of your results?

2.      The age range of the study population is between 6-15 years. Generally, the development of adolescents is more accelerated than that of the children. Could you explain it a bit more somewhere please?

3.      The description of data analysis is not complete. You need to add more for it. E.g in table 4, you wrote the test value as the footnote of the table, although you did not mention the name of the test. In addition, in the table 5, you reported Ä and “B”, but you did not described it in the methodology.

4.      The title is the assessment of nutritional status, however, the authors reported only the information related to body composition throughout the manuscript. Many important variables concerning the nutritional status such as diet, physical activity, lifestyle, etc were missing. Therefore, the results of the manuscript is just reported as descriptive data alone for body composition of school children. Thus, this deviates the objective and title of the study.

5.       Continuation with above comment, the discussion could not be reported thoroughly as a consequence of it.

6.      How about strength and weakness of your study?

7.      How is the generalization of your results?

Minor comments

1.      The information should be added more in the introduction.

2.      The text in line 277-285 are not related to the conclusion.

Author Response

Dear Reviewer,

We are extremely grateful for your very valuable and pertinent suggestions, which enabled us to see our mistakes, especially in the static elaboration of the results, and to reanalyze the data correctly. We respond to comments and suggestions in the attached Word file.

Reviewer 2 Report

General comments

1.       The stated objective of this study, which was to use BIA to assess body composition in children and adolescents with obesity does not align with the stated conclusions of the study. If the purpose of the study is to demonstrate the utility of BIA, then the discussion should discuss how the results reflect the merits of using BIA, relative to other methods.

2.       It appears that the authors may find the general convention of using BMI percentile flawed, but they do not explicitly state their opposition to using BMI percentile. Without such an argument, I find the methods erroneous, as BMI is not considered appropriate for the classification of children and adolescents.

3.       Given the age range of the participants, analyses should adjust for pubertal development through indicators such as Tanner stage. Particularly for those 12 -15, body composition or nutritive status between boys and girls should be examined separately. The authors should examine whether they are powered to perform such analyses.

4.       The formatting for tables in the manuscript is difficult to interpret. Consider revising.

5.       Please describe any pre-measurement conditions that were controlled such as clothing,  fasting/fed state, physical activity, voiding bladder.

Minor comments

1.       Abbreviations BMI, WHR, and WHtR are not defined in the manuscript. Please define at first use.

2.       In the results section the authors indicate the highest and lowest values for several variables. It would be more concise to list these values as mean ± standard deviation [ range] .

3.       In the introduction the authors mention that BIA can be used for the assessment of “bone tissue”. Given that bone is not conductive, BIA does not assess bone. Some analyzers may include estimates, but given that this method is not designed to assess bone, please remove this statement as it is misleading, particularly to the novice BIA user. 

Author Response

(The authors gave the same response as above.)

Reviewer 3 Report

The authors report on obese and overweight prevalence in Polish children. The findings appear important but it is unclear if the data currently existed and how this study contributes new information.

Comments:

Abbreviations such as BIA and ANOVA should be defined once on first use and then abbreviated throughout the paper.

"Percentile" is preferred to "centile."

"No statistical significance" should be restated as "no statistically significant effect/difference..."

The need for the study is not described in the introduction. What data existed on childhood body weight prior to the study? Were data lacking? Readers should be convinced of the novel contribution of your findings to the evidence base.

Line 48: Define each abbreviation

Table 3: "No abdominal" and "abdominal" are unclear phrases. Make these more understandable.

Tables 3 & 4 need more descriptive titles and explanatory footnotes.

Table 5: Not all values have a letter showing differences among means.

Lines 170-171: Include reference/citation.

Line 185: in THE case

Line 280:...create eating patterns

Author Response

(The authors gave the same response as above.)

Round 2

Reviewer 1 Report

Thank you for your interesting manuscript! However, I still have some suggestions.

1.      Regarding with the sample size, how did you fix the effects of small sample size of the study such as validity of the results, type II errors, etc as far as the sample size is relatively small. Otherwise, you can explain it sample size if you have already checked for it to be enough.

2.      You could discuss the body composition of the adolescents in relation to diet pattern, physical activity or sedentary lifestyle.  

3.      The strength and weakness of the study is still missing.

4.      The generalization of the results must be reported in the manuscript.

Author Response

Dear Reviewer,

We are extremely grateful for your very valuable and pertinent suggestions. We respond to all comments and suggestions in the attached file.

Reviewer 2 Report

The author's responses and revisions have improved the manuscript significantly. However, the manuscript needs further revision before publication. 

Overall, it is unclear what the objective, hypotheses, and purpose of this manuscript are.  The statement of purpose in the introduction is vague and no hypothesis is given. Conversely, the discussion is written as if the article's objective was to determine the validity of BMI in this group. Such inferences cannot be made from this analysis.  

The authors should state why this manuscript is needed, and what contribution it makes to the literature, as it is currently unclear. 

Below are general comments

* The title is unnecessarily wordy. Maybe reduce the title to "Body composition and anthropometric indicators in children and adolescents 6-15 years old".  This could be further reduced by removing "anthropometric indicators, as this is encompassed by body composition. 

* page 2 , line 65 remove electrical from before bioimpedance. 

* page 3 , lines 117, 118, and 119.  The "÷ "  symbol should not be used here because it indicates division and may confuse the reader .

* the data analysis section should be revised to improve clarity.  As written it is difficult to follow and unclear why these analyses are being performed. 

* overall the line and paragraph spacing should be checked for consistency

* make sure that the tables don't get cut off from their headers because it is impossible to follow as they are in the current draft.

* In table 1, does "medium" represent the mathematical median? If so, consider changing to median. Also, it may be more appropriate to change this to the mean to be consistent with the rest of the manuscript. 

* In the conclusion, revise this section to match the purpose of the manuscript. Consider removing the statements on line 367 and 368, as your analysis does not support this claim, however, factual it might be. 

Author Response

(The authors gave the same response as above.)

Reviewer 3 Report

The authors have addressed my comments. Below are listed minor suggestions.

Table 1. "Medium" is not the proper stats term.

Table 4. Define "M"

BMI index is redundant.

Line 23: BMI

Line 361: are not

Author Response

(The authors gave the same response as above.)
